Schizoperavietnamica sp. nov. (Copepoda, Harpacticoida, Miraciidae), a new species from a mangrove zone in Vietnam, with description of a new species, and a key to the species from the Oriental region, Sulawesi, and East Asia

Tran Ngoc-Son 1
Pham Thi-Phuong 1
Dam Minh Anh 1
Boonyanusith Chaichat chaichat.b@nrru.ac.th 2
1 The University of Danang - University of Science and Education , Danang City , Vietnam
2 School of Biology, Faculty of Science and Technology, Nakhon Ratchasima Rajabhat University , Nakhon Ratchasima , Thailand
Idris Izwandy
Electronic publication date: 2025 Oct 31
Publication date: 2025
Volume: 13
Electronic Location ID: e20246
Received 2025 Jun 28; Accepted 2025 Sep 25
Copyright: ©2025 Tran et al.
Copyright year: 2025
Copyright holder: Tran et al.
License: This is an open access article distributed under the terms of the Creative Commons Attribution License, which permits unrestricted use, distribution, reproduction and adaptation in any medium and for any purpose provided that it is properly attributed. For attribution, the original author(s), title, publication source (PeerJ) and either DOI or URL of the article must be cited.
License URL: https://creativecommons.org/licenses/by/4.0/

Keywords: Brackish water, Coastal environment, Southeast Asia, West Pacific, Taxonomy, Biodiversity

Funding: Ministry of Education and Training, Vietnam B2023.DNA.16 This work was financially supported by Funds from the Ministry of Education and Training, Vietnam; grant number B2023.DNA.16. The funders had no role in study design, data collection and analysis, decision to publish, or preparation of the manuscript.

==============================
Background

A new species of harpacticoid copepod of the genus Schizopera, S. vietnamica sp. nov., was found. This is the first record of the genus for the country.

Methods

Samples were collected in July 2023, using a plankton net with a mesh size of 50 µm. The body parts of the specimens were dissected and mounted on glass slides. Morphological examination was carried out at 1,000× magnification. Habitus and appendages were then drawn using a drawing tube attached to a compound microscope.

Results

The new species differed from all its congeners in the relative length and chaetotaxy of the caudal rami, length of the proximal endopodal segment of the female first swimming leg relative to the exopod, armature of the middle and the distal endopodal segments of the second to the fourth swimming leg, segmentation and chaetotaxy of the fifth swimming leg. The new species resembles most S. neglecta in the loss of the inner seta on the proximal endopodal segment of the second to the fourth swimming legs, as well as in the shape and setular ornamentation of the inner margin of the caudal rami. A key to the species of the genus known from the Oriental region, Sulawesi and East Asia is provided.

Introduction

Research interest in biodiversity patterns of Copepoda has increased rapidly in Vietnam, resulting in the discovery of many new copepod taxa, primarily from subterranean environments. At present, a total of 18 species have been discovered and described from karstic and alluvial aquifers (Borutzky, 1967; Dang & Ho, 2001; Brancelj, 2005; Tran & Chang, 2012; Kołaczyński, 2015; Tran & Hołyńska, 2015; Tran & Brancelj, 2017; Sanoamuang, Boonyanusith & Brancelj, 2019; Tran & Chang, 2020; Tran, Trinh-Dang & Brancelj, 2021; Tran et al., 2025; Tran, Nguyen & Brancelj, 2025; Tran, Phung & Watiroyram, 2025). Among these, four new species of the calanoid families Pseudocyclopidae Giesbrecht, 1893 (one species) and Diaptomidae Baird, 1850 (three species), as well as four new cyclopoids of the family Cyclopidae Rafinesque, 1815, have been described. Notably, all newly discovered Diaptomidae and two Cyclopidae species were assigned to three newly established genera restricted to karst water bodies of Vietnam: Nannodiaptomus Dang & Ho, 2001, Hadodiaptomus Brancelj, 2005, and Pseudograeteriella Sanoamuang, Boonyanusith & Brancelj, 2019. In addition, a new record of Bryocyclops anninae Menzel, 1926 (Cyclopoida: Cyclopidae) from Thien Duong Cave in Central Vietnam has been documented (Sanoamuang, Boonyanusith & Brancelj, 2019).

The remaining ten newly described taxa belong to Harpacticoida, and represent the families Canthocamptidae Brady, 1880 (three species), Tachidiidae Sars G.O., 1909 (one species), Ameiridae Boeck, 1865 (two species), Parastenocarididae Chappuis, 1940 (two species), and Phyllognathopodidae Gurney, 1932 (two species) (Tran & Chang, 2012; Tran, Trinh-Dang & Brancelj, 2021; Tran, Nguyen & Brancelj, 2025; Tran, Nguyen & Brancelj, 2025; Tran, Phung & Watiroyram, 2025). This is particularly significant as no new species of Miraciidae Dana, 1846 has been documented in Vietnam, despite the existence of brackish-water habitats covering 3,069 km2, which represents 0.96% of the national territory (Murray et al., 2019). Thus, these findings suggest that copepod biodiversity in brackish water environments of Vietnam remains largely understudied.

Schizopera Sars, 1905 is the most diverse taxon of the family Miraciidae (Sönmez, Sak & Karaytuğ, 2015; Walter & Boxshall, 2025), The genus was established by Sars (1905) to accommodate S. longicauda Sars, 1905 from the Chatham Islands, about 500 nautical miles east of New Zealand (Sars, 1905). Wells (2007) listed 85 valid species and subspecies, excluding S. rybnikovi Chertoprud & Kornev, 2005. Then, 18 species and subspecies were subsequently described from Uzbekistan (two species) (Mirabdullayev & Ginatullina, 2007), Australia (eight species and subspecies) (Karanovic & Cooper, 2012), Colombia (one species) (Fuentes-Reinés & Gómez, 2014), Japan (one species) (Karanovic, Kim & Grygier, 2015), Turkey (one species) (Sönmez, Sak & Karaytuğ, 2015), Korea (four species) (Karanovic & Cho, 2016) and Thailand (one species) (Watiroyram, Sanoamuang & Brancelj, 2021). Nevertheless, S. spinulosa Mirabdullayev & Ginatullina, 2007 is not accepted, having been recognized as a junior homonym of S. spinulosa Sars, 1909 (Karanovic & Cooper, 2012; Walter & Boxshall, 2025). Thus, 103 species and subspecies of the genus have been validated (Walter & Boxshall, 2025).

The genus Schizopera appears to be cosmopolitan, having been frequently encountered in coastal marine environments worldwide, including the marine littoral, estuaries, coastal lagoons, and the interstitial waters of beaches (Karanovic, 2006). Many species also occupy either brackish or freshwater environments (Sönmez, Sak & Karaytuğ, 2015), of which two species flocks have been recognized so far from Lake Tanganyika (Sars, 1909; Gurney, 1928; Lang, 1948; Rouch & Chappuis, 1960) and Western Australia (Karanovic, 2004; Karanovic, 2006; Karanovic & Cooper, 2012; Karanovic & McRae, 2013). Only 13 species and subspecies have been recorded from the Oriental zoogeographical region and Sulawesi, mostly in India, Indonesia and Thailand (Table 1). However, the genus has never been recorded in Vietnam.

During investigation on the copepods from the lower reaches of central Vietnam, a new species of the genus Schizopera, S. vietnamica sp. nov., was discovered, representing the first record of the genus for the country. The morphological description and illustration of the new species are provided, and a key to species of the genus known from the Oriental region, Sulawesi, and East Asia is proposed.

Table 1 List of Schizopera recorded in Oriental zoogeographical region and Sulawesi.

Taxa	Localities	References	
S. clandestina (Klie, 1923)	Guangdong (China)	Chertoprud, Gómez & Gheerardyn (2009); Shen et al. (1979)	
S. consimilis Sars, 1909	Goutami-Godavari estuary (Andhra Pradesh, India)	Dev Roy & Venkataraman (2018)	
S. crassispinata Chappuis, 1954	Waltair (Andhra Pradesh, India)	Dev Roy & Venkataraman (2018)	
S. indica Rao & Ganapati, 1969	Lawson’s Bay (Andhra Pradesh, India)	Rao & Ganapati (1969)	
S. knabeni Lang, 1965	Peninsular of Malaysia	Shafie & Rahim (2021)	
S. longirostris (Daday, 1901)	Wat Sabatome (Bangkak)Gulf of Thailand (Thailand)	Daday (1906); Chertoprud, Gómez & Gheerardyn (2009); Maiphae & Sa-ardrit (2011)	
S. monardi Petkovski, 1955	Goutami Godavari estuarine system (Andhra Pradesh, India)	Dev Roy & Venkataraman (2018)	
S. neglecta Akatova, 1935	Xiamen (Fijian); Jiangsu; Tianjin (China)	Shen et al. (1979)	
S. paktaii Watiroyram, Sanoamuang & Brancelj, 2021	Cave, about 28 km from Andaman Sea (Thailand)	Watiroyram, Sanoamuang & Brancelj (2021)	
S. spinifer Wells & Rao, 1987	Long Island (Andaman Islands)	Wells & Rao (1987)	
S. subterranea Lang, 1948	Gulf of Thailand	Maiphae & Sa-ardrit (2011)	
S. tobae tobae Chappuis, 1931	Toba Lake (Sumatra); freshwater bodies (West Java)	Chappuis (1931)	
S. tobae wolterecki Brehm & Chappuis, 1935	Lake Towuti (Sulawesi)	Brehm & Chappuis (1935)	

Materials & Methods

Twelve samples were collected through horizontal tow from the mangrove area of the Vu Gia–Thu Bon River in Quang Nam Province, central Vietnam, in July 2023, using a plankton net with a mesh size of 50 µm. All materials were immediately fixed in 70% ethanol. Once in the laboratory, specimens were sorted under a Stemi 508 Carl Zeiss stereomicroscope at 40× magnification; a few of them were then dissected and mounted in a drop of pure glycerol on glass slides sealed with transparent nail varnish. The remaining whole specimens were retained in 70% ethanol for reference. The examination of body parts and ornamentation were carried out at 1,000× magnification under a Axio Lab A1 Carl Zeiss compound microscope. Habitus and appendages were then drawn using a drawing tube attached to a compound microscope at 400× and 1,000× magnifications, respectively.

The morphological description follows the terminology of Huys & Boxshall (1991). Abbreviations employed in the description are: ae, aesthetasc; Enp, endopod; Exp, exopod; Exp/Enp-1(-2, -3), proximal (middle, distal) segment of exopod or endopod; P1–P6, first to sixth swimming legs. For the armature formula of the antennule, Roman numerals denote the segments, and Arabic numerals denote the number of elements on each segment.

The type materials are deposited at the Zoological Collection of Duy Tan University (ZC-DTU), Da Nang city, Vietnam.

The electronic version of the published work and nomenclatural acts have been registered in ZooBank. By putting the LSID (Life Science Identifier) behind the prefix http://zoobank.org/, the ZooBank LSID can be resolved and the associated information viewed through any standard web browser. The LSID for this publication is: urn:lsid:zoobank.org:pub:50CEF9AE-6C53-4366-BBD8-98ABE5E2F9FD. The online version of this work is archived and available from the following digital repositories: PubMed Central, Zenodo and CLOCKSS.

Results

Systematics

Order Harpacticoida Sars, 1903	
Family Miraciidae Dana, 1846	
Subfamily Diosaccinae Sars, 1906	
Genus SchizoperaSars, 1905	
Schizopera vietnamica sp. nov.	
urn:lsid:zoobank.org:act:FEC551E5-DBCA-4370-B0F7-347EDB0A2D76	
Figs. 1–5 (female); Figs. 6–8 (male)	

Material examined. Adult female holotype (ZC-DTU-COPEPODA-0015), completely dissected and mounted on a slide in glycerol and sealed with nail vanish, collected from the type locality, 10 July 2023. Adult male allotype, collection data as for holotype (ZC-DTU-COPEPODA-0016). Paratypes, three adult females and three adult males, collection data as for holotype (ZC-DTU-COPEPODA-0017).

Figure 1 Schizopera vietnamica. sp. nov. female, holotype (ZC-DTU-COPEPODA-0015).

(A) Habitus, dorsal view. (B) Genital double-somite, urosomites 4–5, and anal somite with caudal rami, lateral view. (C) Genital double-somite, urosomites 4–5, and anal somite with caudal rami, dorsal view. (D) Genital double-somite, urosomites 4–5, and anal somite with caudal rami, ventral view.

Diagnosis. Female: body moderately cylindrical. Rostrum distinct. Caudal ramus ca. 1.4 times as long as wide, with setules on inner margin. Distal exopodal segment of the antenna with three apical elements. Coxa of the maxillule with one apical spine. Exp and Enp of all swimming legs three-segmented. P1Enp-1 reaching tip of Exp-3, with inner seta. Exp-1, Exp-2, and Enp-1 of P2–P4 without inner seta. Armature complement of Exp-3 and Enp-3 of P1–P4: 4.4.4.4 and 3.4.4.3, respectively. Left and right legs of P5 separated; P5Exp suboval, with six elements; endopodal lobe of baseoendopods reaching mid of Exp, with four setae. P6 reduced to a small protuberance with two apical setae on peduncle. Male: P2 with Enp-2 and Enp-3 completely incorporated, with outer subapical seta transformed. P3 with hyaline spine reaching distal 2/3 of distal half of Exp-3. P5 baseoendopods fused medially, with five setae on Exp and two subequal marginal setae on endopodal lobe. P6 reduced to short cuticular bilobate plates, unarmed.

Type locality. The mangrove area of the Vu Gia–Thu Bon River, Quang Nam Province, Vietnam (15°52′31.0″N 108°22′35.0″E).

Etymology. The specific epithet is derived from Vietnam, alluding to the name of the country where the new species was discovered. The name is an adjective in the nominative singular, gender feminine.

Description of adult female. Total body length, measured from tip of rostrum to posterior margin of caudal rami: 370–434 µm (mean = 408 µm, n = 4), 410 µm in holotype. Habitus cylindrical, slightly tapering posteriorly (Fig. 1A); preserved specimens colourless. Prosome comprising cephalothorax and three free pedigerous somites; urosome comprising fifth pedigerous somite, genital double-somite (fused genital and first abdominal somites) and three free abdominal somites; prosome/urosome ratio about 1.4 (in dorsal view).

Rostrum distinct, relatively long, triangular with round tip, reaching distal 2/3 of segment II of antennule, with pair of sensilla at distal 2/3 (Fig. 1A).

Cephalothorax about 1.4 times as long as wide, with a single red nauplius eye in the middle (not shown) and sensillae as shown in Fig. 1A; hyaline frill with smooth margin. Free pedigerous somites with continuous rows of minute spinules on dorsal view and sensillae near posterior margin; hyaline frill on all somite of prosome with smooth margin.

Urosomites 1–4 dorsally with numerous rows of tiny spinules and serrated hyaline frill dorso-laterally and ventrally; surface of urosomites 2–4 with pairs of cuticular pores and sensillae on each somite as shown (Figs. 1A–1D). Genital double-somite (Figs. 1A, 1C, and 1D) as long as wide; genital complex (Fig. 1D) with one copulatory pore posterior to epicopulatory bulb and two small seminal receptacles on both sides. Anal somite (Figs. 1A–1D) with rows of tiny spinules dorso-laterally, and with ventrally and row of larger spinules on distal margin; inner margin with oblique row of long setules posterior to anal operculum. Anal operculum (Figs. 1A and 1C) crescentic, free margin with slim and short setules.

Caudal rami (Figs. 1C and 1D) slightly divergent, tapering posteriorly, about 1.4 times as long as wide; ornamented with row of small spinules and one cuticular pore on dorsal view, with long median setules along the inner margin, and a row of ventral spinules on distal margin. Anterolateral accessory seta (I) absent. Anterolateral seta (II) robust, spiniform, about 0.6 times as long as ramus, inserted at distal 2/3. Posterolateral seta (III) smooth, relatively slender, about 0.9 times as long as ramus, inserted near base of seta II. Outer apical seta (IV) smooth, shorter than inner apical seta (V), with breaking plane. Inner apical seta (V) longest, bipinnate, about 4.5 times as long as ramus, with breaking plane. Inner accessory seta (VI) bare, about 0.4 times as long as ramus. Dorsal seta (VII) triarticulated, slender, plumose at tip; located near mid-length of ramus.

Antennule (Fig. 2A) eight-segmented, about half as long as cephalothorax. Aesthetasc on segment 4 well developed, reaching tip of segment 8 at about the middle of its length; aesthetasc on segment 8 thin and short. Aesthetasc on segment 4 and 8 fused with two neighbouring setae at base, forming acrothek. Setal formula as follows: I-[1], II-[7], III-[6], IV-[1+acrothek], V-[1], VI-[3], VII-[3], VIII-[4+acrothek].

Figure 2 Schizopera vietnamica sp. nov. female, holotype (ZC-DTU-COPEPODA-0015).

(A) Antennule. (B) Antenna. (C) Mandible. (D) Maxillule. (E) Maxilla. (F) Maxilliped.

Antenna (Fig. 2B) comprising coxa, basis, two-segmented Exp and one-segmented Enp. Coxa unornamented. Allobasis about 2.1 times as long as wide, with several short spinules on posterior and ventral views, with abexopodal unipinnate seta margin. Exp-1 with one pinnate subapical seta on inner distal corner; Exp-2 with three apical elements: one minute, smooth inner seta, one bipinnate spine and one thin, bare seta of about same length as spine. Enp about 2.5 times as long as wide; ornamented with spinules along medial margin; armed with two robust spines and one thin, smooth seta on medial margin, and seven apical elements: one geniculate pinnate seta fused with one geniculate seta, three geniculate bare apical setae, one spiniform smooth apical seta on inner corner, and one slender, smooth seta near the first mentioned geniculate one.

Mandible (Fig. 2C) with ten strong chitinized teeth and one unipinnate dorsal seta on coxal gnathobase. Basis with three pinnate setae unequal in length inserted on inner margin. Enp one-segmented, with two lateral, and two apical pairs of setae fused basally. Exp one-segmented, very small, with one smooth apical seta.

Maxillule (Fig. 2D) comprising robust praecoxa, coxa, basis, Exp and Enp. Praecoxal arthrite with several spinules laterally; armed with eight strong spines on distal margin (six bare and two pinnate spines) and two surface setae. Coxa small, endite with one strong, curved, bare spine. Basis with one spiniform pinnate seta, two smooth and one slender, pinnate seta; ornamented with spinules along outer margin. Exp small, one-segmented; armed with two bare apical setae. Enp one-segmented, about twice as long as wide, with three smooth apical setae of unequal length.

Maxilla (Fig. 2E) consists of syncoxa, allobasis, and one-segmented Enp. Syncoxa with three endites; proximal and middle endites with one unipinnate, and one bare apical seta; distal endite with two smooth apical setae. Basis drawn-out into strong claw, unilaterally pinnate distally, with two thin, smooth setae near base. Enp small, one-segmented, armed with two bare apical setae.

Maxilliped (Fig. 2F) prehensile. Syncoxa 1.3 times as long as wide, with two unipinnate setae and one pinnate seta on inner distal corner. Enp two-segmented; Enp-1 about 2.3 times as long as wide, ornamented with a row of strong spinules near inner margin and a row of long spinules on outer margin, armed with two smooth setae on inner margin; Enp-2 with four apical elements: one claw-like, smooth spine and three short, smooth setae of subequal length.

P1–P4 (Figs. 3 and 4 and 5A) with three-segmented rami. Coxae in all swimming legs connected by unornamented intercoxal plate; P2–P4 with intercoxal plates with acute projections on distal margin. Armature formula of P1–P4 as in Table 2.

Figure 3 Schizopera vietnamica sp. nov. female, holotype (ZC-DTU-COPEPODA-0015).

(A) P1. (B) P2.

Figure 4 Schizopera vietnamica sp. nov. female.

(A) P3, holotype (ZC-DTU-COPEPODA-0015). (B) Left P3, paratype (ZC-DTU-COPEPODA-0017).

P1 (Fig. 3A). Coxa with a row of spinules on outer margin. Basis with one inner and one outer pinnate spine. Exp three-segmented, shorter than Enp, with spinules along outer margin in all segments and spinules on distal margin in Exp-1 and Exp-2; Exp-1 with additional row of spinules on anterior surface, with one strong unipinnate outer spine; Exp-2 with one robust unipinnate outer spine; Exp-3 with four elements: two smooth outer spines and two geniculate unipinnate apical setae. Enp three-segmented; Enp-1 about 3.7 times as long as wide, reaching tip of Exp-3, with row of spinules along inner and outer margin, with one robust, smooth seta inserted at distal 2/3 of Enp-1; Enp-2 relatively short, with several spinules along outer margin and unarmed; Enp-3 about 1.6 × as long as wide and about 1.5 times as long as Enp-2, with three elements: one slim, bare inner seta, one inner, geniculate and pinnate, apical seta, and one outer, unipinnate, apical spine.

P2 (Fig. 3B). Coxa with several spinules on anterior surface and near outer margin. Basis with a row of spinules on distal margin, with one strong unipinnate spine on outer margin. Exp with spinules and setulae along outer and inner margin, respectively, in all segments, and with spinules on inner distal corner in Exp-1 and Exp-2; Exp-1 with additional row of strong spinules on anterior surface, with one robust pinnate outer spine; Exp-2 with one robust pinnate outer spine; Exp-3 with four elements: two pinnate apical setae, one pinnate outer spine subapically, and one unipinnate outer spine. Enp reaching tip of Exp-3, Enp-1 and Enp-2 with spinules and setulae along outer and inner margin, respectively; Enp-1 about 1.5 times as long as wide, with additional row of spinules on inner distal corner, unarmed; Enp-2 with one unipinnate inner seta inserted at distal 2/3 of the segment, with additional row of spinules on inner distal corner; Enp-3 unornamented, with one unipinnate inner seta, two pinnate apical setae, and one pinnate outer spine subapically.

P3 (Fig. 4A). Coxa as in P2. Basis with a row of spinules on distal margin, and a thin, smooth outer seta. Exp as in P2, except for the absence of setules on inner margin of Exp-1. Enp reaching distal 2/3 of Exp-3, with spinules along outer margin in all segments and spinules along inner margin in Enp-1 and Enp-2, with additional row of spinules on inner distal corner of Enp-2; Enp-1 about 1.4 times as long as wide, unarmed; Enp-2 with one plumose inner seta; Enp-3 as long as Enp-2, with a row of spinules along outer margin, and armed with four elements: one robust unipinnate inner seta, two pinnate apical setae, one pinnate outer spine subapically.

P4 (Fig. 5A). Coxa unornamented. Basis with one bare outer seta. Exp as in P2 and P3. Enp reaching proximal 1/3 of Exp-3, and ornamentation as in P3, except Enp-3 with additional row of spinules on inner margin; Enp-1 unarmed; Enp-2 with one plumose inner seta; Enp-3 with three elements: two pinnate apical setae, one pinnate outer spine subapically.

P5 (Fig. 5B) with distinct Exp and baseoendopod. Exp subcircular, as long as wide; with six elements: five pinnate setae and one slender smooth seta, of which the fourth and the fifth (from inner to outer) shortest, subequal in length, about 1/3 of the outermost seta. Baseoendopod well-developed, reaching middle of Exp; with several spinules on the outer margin of endopodal lobe; with thin bare outer seta and four marginal spiniform elements, of which the third (from inner to outer) longest, innermost seta shortest.

P6 (Fig. 1D) reduced to minute prominence and fused to somite bearing it, with one plumose outer seta and one longer smooth inner seta on peduncle.

Description of adult male. Total body length, measured from tip of rostrum to posterior margin of caudal rami: 364–394 µm (mean = 375 µm, n = 4), 380 µm in allotype, slightly smaller than female. Habitus, ornamentation, colour and nauplius eye as in female except for sexually dimorphic urosome with six somites (genital somite and first abdominal one not fused) (Figs. 6A–6D).

Antennule (Fig. 7A) sexually dimorphic, eight-segmented; approximately half as long as cephalothorax. Aesthetasc on segment 4 well developed, reaching tip of segment 8 at about the middle of its length and fused with neighbouring seta at base; that of segment 8 relatively shorter than in the female and fused with two neighbouring setae at base, forming acrothek. Setal formula as follows: I-[1], II-[6], III-[3], IV-[12+ae], V-[1], VI-[1], VII-[4], VIII-[4+acrothek].

Figure 5 Schizopera vietnamica sp. nov. female, holotype (ZC-DTU-COPEPODA-0015).

(A) P4. (B) P5.

Figure 6 Schizopera vietnamica sp. nov. male, (A) paratype (ZC-DTU-COPEPODA-00 17) and (B–D) allotype (ZC-DTU-COPEPODA-00 16).

(A) Habitus, dorsal view. (B) Genital double-somite, urosomites 3–5, and anal somite with caudal rami, lateral view. (C) Genital double-somite, urosomites 3–5, and anal somite with caudal rami, dorsal view. (D) Genital double-somite, urosomites 3–5, and anal somite with caudal rami, ventral view.

Figure 7 Schizopera vietnamica sp. nov. male, allotype (ZC-DTU-COPEPODA-0016).

(A) Antennule; (B) P1; (C) P5.

P1 (Fig. 7B). Coxa ornamented with row spinules along outer margin. Basis with row of spinules on outer and distal margin; with one pinnate outer spine and obtuse inner spine less sharp than in female. Exp and Enp as in female.

P2 (Fig. 8A) sexually dimorphic. Coxa with several spinules on anterior surface. Basis with robust unipinnate outer spine and a row of tiny spinules on distal margin. Exp three-segmented, as in female. Enp two-segmented; Enp-1 about 1.5 times as long as wide, with a row of setules along inner margin. Enp-2 and Enp-3 completely fused, and with outer distal corner produced into long, blunt spiniform process slightly longer than transformed outer apical spine; with one smooth inner seta, and three apical elements: one outer subapical seta transformed into smooth spiniform element with spike at tip, one inner subapical seta slender and smooth, subequal in length to the outer one, and one apical seta elongate and unipinnate on the distal half.

Figure 8 Schizopera vietnamica sp. nov. male, allotype (ZC-DTU-COPEPODA-0016).

(A) P2. (B) P3.

P3 (Fig. 8B) sexually dimorphic, armed as in the female except for the anterior surface of Exp-3 with enlarged tubular pore inserted at the middle and close to inner margin, reaching distal 2/3 of distal half of Exp-3.

P4 (not shown) as in female.

P5 (Fig. 7C) with Exp and baseoendopod separated. Exp subcircular, as long as wide, with five marginal elements of which the second from inner to outer longest and pinnate, and the fourth shortest and smooth; length ratio of marginal setae from the innermost first to the outermost fifth: 1: 2.5: 2.2: 0.6: 1.2. Each both baseoendopods fused medially forming a single plate, with a row of spinules on outer margin, with one thin, bare outer basal seta, and two robust endopodal pinnate setae of unequal length.

P6 (Fig. 6D) represented by a pair of small and short, unarmed and unornamented plates.

Variability. While the inner seta is absent in P3Enp-1 of holotype, a female paratype has one pinnate inner seta on the left P3Enp-1 (Fig. 4B).

Discussion

Differential diagnosis and remarks

According to Lang (1965), the new species belongs to the genus Schizopera, because it exhibits a combination of the following characteristics:

(1) The presence of a hyaline spine on the male P3Enp-3,

(2) Uniform transformation of the inner spine on the male P1 basis,

(3) Characteristic features of the female genital area,

(4) Loss of the proximal outer spine on the Exp-3 of P1–P4,

(5) Antenna with allobasis and two-segmented Exp.

The combination of these characteristics has been proposed as the evidence of the monophyletic relatedness of the genus (Gómez & Vargas-Arriaga, 2008), although the presence of the hyaline spine is not specific to it, having been found in Eoschizopera Wells & Rao, 1976, too (Wells & Rao, 1976). Some authors have defined the spine as an enlarged tubular pore (e.g., Karanovic, 2006; Karanovic & McRae, 2013; Karanovic, Kim & Grygier, 2015). To date, 103 species and subspecies have been validated, excluding the new species presented herein (Walter & Boxshall, 2025). Among 103 valid taxa, the loss of inner seta in both the Exp-2 and the Enp-1 of P2–P4 in the new species is shared by only four species. They are S. bozici Lang, 1965 reported from California (Lang, 1965) and the Saint-Philippe of the La Réunion (approximately 750 km eastward of Madagascar) (Bozic, 1964; Bozic, 1969); S. noodti Rouch, 1962 described from Buenos Aires, Argentina (Rouch, 1962); S. akolos Karanovic & Cooper, 2012 from the Yilgarn region, Western Australia (Karanovic & Cooper, 2012); and S. gangneungensis Karanovic & Cho, 2016 from Korea (Karanovic & Cho, 2016). The new species considerably differs from them in the relative length and the chaetotaxy of caudal rami, the length of the female P1Enp-1 relative to Exp, the armature complement of Enp-2 and Enp-3 of P2–P4, and segmentation and chaetotaxy of P5 (Table 3). Among them, the Vietnamese Schizopera most resembles S. noodti in the segmentation and the armature of P5, the length of P1Enp-1 relative to Exp, and the armature complement of Enp-2 of P2–P4. However, the new species is easily distinguishable from S. noodti based on the following characteristics: (1) the length/width ratio of caudal rami is about 1.4 in the new species, but approximately 2.2 in S. noodti; (2) the setal formula of Enp-3 of P2–P4 is 4.4.3 in the new species, but 4.3.2 in S. noodti (Table 3).

Table 2 Armature formula of female P1–P4 (legend: inner-outer seta/spine; inner-apical-outer seta/spine; Arabic numerals represent number of setae.

Roman numerals represent number of spines).

Leg	Basis	Exp	Enp	
		1	2	3	1	2	3	
P1	I-I	0-I	0-I	0-2-II	1-0	0-0	1-1,I-0	
P2	0-I	0-I	0-I	0-2-II	0-0	1-0	1-2-I	
P3	0-1	0-I	0-I	0-2-II	0/1-0	1-0	1-2-I	
P4	0-1	0-I	0-I	0-2-II	0-0	1-0	0-2-I	

Table 3 Morphological differences among five species of Schizopera without inner seta on both the Exp-2 and the Enp-1 of P2–P4.

Characters	S. vietnamica sp. nov.	S. bozici	S. noodti	S. akolos	S. gangneungensis	
Female						
Shape and length/width ratio of caudal rami	Conical/∼1.4	Conical/∼1.5	Conical/∼2.2	Conical/∼1.0	Conical/∼1.4	
Ornamentation on inner margin of caudal rami	Setules along inner margin	Bare	Setules along inner margin	Transverse row of spinules	Longitudinal row of spinules	
Shape of caudal seta II	Spiniform	Setiform	Spiniform	Spiniform	Spiniform	
Number of elements on distal segments of Exp of antenna and relative length	2 subequal elements	2 subequal elements	1 long and 2 short setae	2 subequal elements	2 subequal elements	
Number of segments of P1Enp	3	3 or 2	3	3	3	
Number of segments of P4Enp	3	3	3	2	3	
Relative length of Enp-1 to Exp of P1	Distal margin of Exp-3	Distal margin of Exp-3	Distal margin of Exp-3	Distal margin of Exp-2	Middle of Exp-3	
Setal formula of Enp-2 of P2–P4	1.1.1	1.0.0 or 1.1.0	1.1.1	1.1.-	1.1.0	
Setal formula of Enp-3 of P2–P4	4.4.3	3.3.3 or 4.3.3	4.3.2	4.2.3	4.2.2	
Segmentation of P5Exp	Free	Fused	Free	Free	Partly fused	
Number of elements on baseoendopod and Exp of P5	4/6	3/5	4/6	4/4	4/5	
Male						
Number of segments of P1Enp	3	2	3	3	NA	

From a geographical viewpoint, the genus is predominantly marine. Most species were discovered in coastal areas around the world. However, only 14 species and subspecies have been reported from the Oriental geographical region and Sulawesi, including new species (Table 1). Of these taxa, S. vietnamica sp. nov. shares the shape and ornamentation of the inner margin of the caudal rami (i.e., the caudal rami are about 1.5 times as long as wide, with setules on the inner margin) with five taxa, which include S. clandestina, S. knabeni, S. longirostris, S. neglecta, S. tobae tobae and S. t. wolterecki, but only S. neglecta lacks the inner seta on Enp-1 of P2–P4, resembling that of the new species. Furthermore, the inner seta on the Enp-1 of P3–P4 is missing in the new species, but it is present on the P3Enp-1 in all the above congeners. In addition, the inner seta on the P4Enp-1 is present in S. clandestina, S. knabeni, S. tobae tobae and S. tobae wolterecki. Because Daday (1901) did not describe or illustrate the P4Enp-1 of S. longirostris, a comparison of the armature complements of the P4Enp-1of that species and the new taxon could not be done.

The new species most resembles S. neglecta described from the Caspian Sea (Akatova, 1935) and later reported from the Black Sea (Jakubisiak, 1938; Monchenko, 1967; Apostolov, 1973), the Mediterranean (Por, 1964), the East China Sea (Shen et al., 1979) and Korea (Chang, 2009), in the loss of the inner seta on Enp-1 of P2–P4, and in the shape and setular ornamentation of the caudal rami. However, the new species is easily distinguishable from S. neglecta by the absence of an inner seta on the Exp-2 of P2–P4, which is present in S. neglecta. Furthermore, the P1Enp-1 reaches the distal margin of P1Exp-3 in the new species rather than just the middle of P1Exp-3 as in S. neglecta.

The loss of the inner seta on Exp-2 of P2–P4 has been recorded in three species from the Oriental region, S. crassispinata described from Madagascar, S. spinifer from India, and S. vietnamica sp. nov. However, the new species is distinguished from S. crassispinata and S. spinifer based on the loss of the inner seta on Enp-1 of P2–P4, and on the difference in the segmentation of P1Enp. Other than S. crassispinata and S. spinifer, the presence of three-segmented P1Enp differentiates S. vietnamica sp. nov. from other 22 species and subspecies of the genus Schizopera in which the P1Enp has two segments (Wells, 2007; Karanovic & Cooper, 2012; Fuentes-Reinés & Gómez, 2014). The number of segments of P1Enp was previously considered by Apostolov (1982) as a main character dividing Schizopera into two subgenera, namely Schizopera and Neoschizopera schizopera, but his subgeneric division scheme was subsequently questioned both based on the morphological (Mielke, 1992; Mielke, 1995) and molecular evidence (Karanovic & Cooper, 2012; Karanovic & McRae, 2013), and has been recently abandoned. An additional character which differentiates new species from S. crassispinata and S. spinifer is the shape of seta on caudal rami and the shape of seta on the baseoendopod of P5, respectively. The caudal seta V of the female is plumose in the new species, but it is transformed to the swollen smooth element in S. crassispinata (Chappuis, 1954: 48, Fig. 10). The two innermost setae on the baseoendopod of the female P5 are the pinnate spiniform elements in the new species, but the two elements are transformed to the pectinate spines in S. spinifer (Wells & Rao, 1987: 323, Fig. 99h).

The loss of the inner seta on Enp-1 of P2–P4 has also been observed in S. consimilis from Lake Tanganyika (Sars, 1909; Wells, 2007), but the new species differs in the relative length of the caudal rami, and P1Enp-1, which is about 1.4 times as long as wide and reaches the distal margin of the Exp in the new species, respectively, whereas it is about twice as long as wide and reaches beyond the distal margin of the Exp in S. consimilis, respectively.

Previously, Watiroyram, Sanoamuang & Brancelj (2021) described S. paktaii from a cave in southern Thailand. The authors considered S. validior Sars, 1909 from Lake Tanganyika (Africa) as the closet relative of the Thai Schizopera. Morphological examination, based on the original description of S. validior, has shown a close affinity between the new species and S. validior according to the similarity of the shape and the ornamentation of the inner margin of caudal rami as well as the armament of the female P5. However, these two species can be easily differentiated by the armature of the Exp-2 and Enp-1 of P2–P4 and by the shape of the P1Enp-3. The inner seta on the Exp-2 and Enp-1 of P2–P4 is lost in the new species, but it is present in S. validior.

Comparing the morphology of the new species and the species not recorded in Wells (2007) has shown that the loss of inner seta in Exp-2 of P2–P4 has not only been observed in S. gangneungensis, but also in S. setulosa (Mirabdullayev & Ginatullina, 2007) from Uzbekistan. Nevertheless, the new species is differentiated from S. setulosa by the loss of the inner seta on Enp-1 of P2–P4 and by the setal formula of Enp-3 of P2–P4 4.4.3 in the new species, but 4.3.2 in S. setulosa.

Diversity pattern

While the number of specialists, who proposed the new species of Copepoda in Vietnam is considered low (15) (see Tran et al., 2025 for the list of taxa and the author of the contribution), the discovery of a large amount of new copepod taxa within a short period might not only indicate the growing interest in the diversity of Copepoda (Boonyanusith, Brancelj & Sanoamuang, 2024) but also in the high diversity of the fauna of Vietnam (Tran et al., 2025). Of 16 newly described copepod species, 13 were discovered and described from the country within the last 13 years.

Following the perspective of Deharveng & Bedos (2000), who utilized supra-specific units (e.g., genus, family, order and so on) to assess differences in diversity patterns across regions, the examination of the list of newly described species in Vietnam reveals a high proportion of copepods that are considered descendants of marine ancestors. Among the 16 newly described species, five belong to the genera Boholina Fosshagen, 1989 in Fosshagen & Iliffe (1989) (Pseudocyclopidae), Microarthridion Lang, 1944 (Tachidiidae), Nitokra Boeck, 1865 (Ameiridae), and Schizopera (Miraciidae) (Tran & Chang, 2012; Tran & Chang, 2020; Tran et al., 2025). They are considered the descendants of marine ancestors because their members or close phylogenetic relatives are predominantly marine or inhabit coastal environments. A similar pattern can be observed in southern and eastern Thailand, where all the newly described taxa in continental water habitats can be classified as freshwater representatives of the marine-ancestral relatives. These include Boholina, Schizopera, Rangabradya Karanovic & Pesce, 2001 (Ectinosomatidae), Onychocamptus Daday, 1903 (Laophontidae), and Cletocamptus Schmankevitsch, 1875 (Canthocamptidae) (Boonyanusith et al., 2018; Boonyanusith, Wongkamhaeng & Athibai, 2020; Boonyanusith & Athibai, 2021; Watiroyram, Sanoamuang & Brancelj, 2021; Boonyanusith & Wongkamhaeng, 2023).

Based on geographical and evolutionary perspectives, some authors have suggested the influence of the proximity to the sea (i.e., the relatively short distance between a given area and the coastline), which significantly contributes to the high species richness of stygobiotic and troglobiotic fauna (i.e., more than 25 obligate subterranean species) in various karst systems, such as the Salukkan Kallang–Tanette Cave System in Indonesia, the Dinaric Karst System in Europe, and the Hon Chong Karst in Vietnam (Deharveng & Bedos, 2012; Zagmajster et al., 2014; Deharveng et al., 2023; Deharveng et al., 2024). This indicates the influence of geographical location on the composition of faunal communities. Regarding Vietnam, the country occupies the easternmost edge of mainland Southeast Asia, stretching approximately 1,600 km from north to south, bordered by about 3,400 km of coastline along the Pacific Ocean and the Gulf of Thailand (Turley et al., 2025). A global remote sensing analysis has identified 3,069 square kilometer of tidal flats, making Vietnam the third-largest intertidal zone in Southeast Asia, after Indonesia and Myanmar (Murray et al., 2019). The geographic features of the country can play a critical role in shaping the high diversity of copepod lineages, many of which possess marine ancestors. Furthermore, brackish environments, which serve as transitional zones between marine and continental systems, are adaptive zones for continental invasions (Lee & Bell, 1999), and have been well documented as a site of radiation and speciation of marine-derived lineages (Lee, 1999). Thus, a high number of Copepoda, that have evolved from marine-ancestral lineages would be expected in Vietnam.

Key to the species and subspecies of Schizopera recorded in the Oriental region, Sulawesi, and East Asia

Throughout the last two decades, 18 species and subspecies of Schizopera were described. Seven species were described from various countries across East and Southeast Asia, including Japan (S. abei Karanovic, Kim & Grygier, 2015), Korea (S. daejinensis Karanovic & Cho, 2016, S. yeonghaensis Karanovic & Cho, 2016, S. gangneungensis Karanovic & Cho, 2016, and S. sindoensis Karanovic & Cho, 2016), Thailand (S. paktaii), and Vietnam (S. vietnamica sp. nov.). These discoveries reflect a growing interest in the biodiversity of Copepoda in the coastal environments of these regions. Accordingly, a key to the species of Schizopera recorded in the Oriental region, Sulawesi, and East Asia is provided below, based primarily on female morphological characteristics. The key also includes S. brusinae Petkovzki, 1954 as Fiers (1986) suggested the species may occur in (at least) the Indian Ocean.

1  – P1Enp two-segmented………….…………….…………………………………….....(2)	
    – P1Enp three-segmented…………….……………………….…………..………..........(5)	
2  – Exp-2 of P2–P4 without inner setae……….….………………………………...…........(3)	
    – Exp-2 of P2–P4 with one inner seta……….……….………………………..……….....(4)	
3  – P5Exp and basesoendopod with five and three marginal setae/spines, respectively; caudal rami about 1.1 times as long as wide, with seta V transformed into swollen smooth element; …………………………………...……....................................... S. crassispinata [Madagascar]	
    – P5Exp and basesoendopod with six and four marginal setae/spines, respectively; caudal rami about 1.2 times as long as wide, with seta V normally developed…......................S. spinifer [Katchal Island (Nicobar Islands)]	
4  – P5Exp with six marginal setae; caudal rami about 1.5 times as long as wide, with short spinule ornamentation on inner margin; setal formulae of Enp-1 and Exp-2 of P2–P4 as 1.0.1 and 0.0.1, respectively ................................ ....S. monardi [the area where is probably in Montenegro]	
    – P5Exp with five marginal setae; caudal rami about twice as long as wide, with long spinule ornamentation on inner margin; setal formulae of Enp-1 and Exp-2 of P2–P4 as 1.1.1 and 0.1.1, respectively; P1Enp-1 reaching mid of Exp-3; caudal seta IV setiform, well surpassing breaking plane of seta V………………………….....................S. brusinae [sensu Kunz (1974), France]	
5  – P4 with two-segmented Enp; caudal rami about twice as long as wide, inner margin naked; caudal seta IV setiform and short, shorter than seta VI; P1Enp-1 reaching tip of Exp-2; setal formula of P2–P4 0.1.1………………………………………........... S.abei [Lake Biwa (Japan)]	
    – P4 with three-segmented Enp……………………………….……….……..……….…(6)	
6  – P5Exp fused to baseoendopod, with five marginal setae; caudal rami with spinule ornamentation on inner margin; setal formulae of Exp-2 and Enp-1 of P2–P4 as 0.0.0…………... …………………………………………………......................... ....S. gangneungensis [Korea]	
    – P5Exp fused to baseoendopod, with six marginal setae………............................…..…..... (7)	
    – P5Exp free, with five marginal setae…………….……………………………...….…....(8)	
    – P5Exp free, with six marginal setae………………………………………….………...(11)	
7  – Exp-2 of P2–P4 without inner setae………………………..... ..........S. daejinensis [Korea]	
    – Exp-2 of P2–P4 with inner setae ..........................................................S. yeonghaensis [Korea]	
8  – Enp-1 of P2–P4 without inner setae; P1Enp-1 surpassing tip of Exp; caudal seta IV setiform, considerably surpassing breaking plane of seta V……….…………………….…….. …………………………………………...... ............S.consimilis [Lake Tanganyika (Africa)]	
    – Enp-1 of P2without inner seta, but Enp-1 of P3 and P4 with inner seta….........................(9)	
9  – Caudal rami about 1.5 times as long as wide, with long spinule ornamentation on inner margin.……………………...... ...…...…............S. brusinae [sensu Fiers (1986), New Guinea]	
    – Caudal rami at least about 1.8 times as long as wide, with setular ornamentation on inner margin……………..……….…………………………..…………... ………………….. (10)	
10  – Caudal seta IV spiniform and short, reaching or slightly surpassing breaking plane of seta V……................S. brusinae [sensu Petkovski (1954), Croatia; sensu Apostolov (1973), Bulgaria]	
    – Caudal seta IV setiform, considerably surpassing breaking plane of seta V……….…..... ………………………………………...……...………………….….....S. paktaii [Thailand]	
11  – All P2, P3 and P4 without inner seta on Enp-1……....………...…….………..….......(12)	
    – Either P2 or P3 and P4 with inner seta on Enp-1…………....………………..…..........(14)	
12  – Exp-2 of P2–P4 without inner seta………….. ..…....... S. vietnamica sp. nov. [Vietnam]	
    – Exp-2 of P2–P4 with inner seta……...…………..……..…..…………………...….....(13)	
13  – Setal formula of Enp-3 of P2–P4 4.4.3………….............……………...……………… ………………S. neglecta [sensu Akatova (1935), Russia; sensu Por (1964), Israel; sensu Shen et al. (1979), China; sensu Chang (2009), Korea; and sensu Apostolov (1973), Bulgaria]	
    – Setal formula of Enp-3 of P2–P4 4.3.3…......S. neglecta [sensu Monchenko (1967), Romania]	
14  – P2Enp-1 with inner seta; P1Enp-1 surpassing tip of Exp…………………………… ...……………………………...………………............ S. longirostris [New Guinea; Thailand]	
    – P2Enp-1 without inner seta, but P3 and P4 with inner seta on Enp-1…………………(15)	
15  – Setal formula of Exp-3 of P2–P4 4.4.5; setal formula of Enp-3 of P2–P4 5.5.3…...…….. ………………………………………………………………….……......... S. indica [India]	
    – Setal formula of Exp-3 of P2–P4 4.4.4 or 4.3.4; setal formula of Enp-3 of P2–P4 3.4.3, 4.4.3 or 4.5.3…………………………………………………………………………...............(16)	
16  – Exp-2 of P2 with inner seta, but Exp-2 of P3 and P4 without inner seta; setal formula of Enp-3 of P2–P4 4.5.3 ……………..…S.clandestina [sensu Arlt (1983), Baltic Sea (German)]	
    – Exp-2 of P2 and P3 without inner seta, but Exp-2 of P4 with inner seta…...…......…...... …………………………………………………………………………S. sindoensis [Korea]	
    – Exp-2 of P2–P4 with inner seta…………….…………….………………………......(17)	
17  – Setal formula of Enp-3 of P2–P4 3.4.3.....S.clandestina [sensu Klie (1923), Germany]	
    – Setal formula of Enp-3 of P2–P4 4.4.3…………....…………….……………..……...(18)	
18  – Caudal rami without setules on inner margin; P1Enp-1 reaching tip of Exp-2…...…… ……………………………..………………………............S. subterranea [Italy; Thailand]	
    – Caudal rami without setules on inner margin; P1Enp-1 reaching tip of Exp-3….........…(19)	
    – Caudal rami with setules on inner margin; P1Enp-1 reaching tip of Exp-2 at least but not surpassing tip of Exp-3……..……………..………………………………..........................(20)	
    – Caudal rami with hairs on inner margin; P1Enp-1 surpassing tip of Exp ...................….. (21)	
19  – Caudal rami about 1.5 times as long as wide …S. tobaetobae [Sumatra, Java (Indonesia)]	
    – Caudal rami about 1.7–1.8 times as long as wide……..……………..………..…………. ……………………………………………………S. tobae wolterecki [Sulawesi (Malaysia)]	
20  – Caudal rami about 1.5–2 times as long as wide……………….................S.clandestina [sensu Apostolov (1973), Bulgaria; sensu Chang (2009), Korea; sensu Shen et al. (1979), China]	
    – Caudal rami as long as wide……………….…......... S.clandestina brevicauda [Germany]	
21  – P1Enp-3 about twice as long as wide.…….….……..............S. knabeni [America; Mexico]	
    – P1Enp-3 about three times as long as wide.....…….….………….….……….……….…… ….……….……….……………………………S. clandestina [sensu Noodt (1953), Germany]	

Conclusions

Schizopera vietnamica sp. nov. was described based on specimens collected from the mangrove zone in the Vu Gia–Thu Bon coastal region, Quang Nam Province, central Vietnam. The species is distinguished by a unique combination of morphological characters, including the relative length and chaetotaxy of the caudal rami, the length of the first endopodal segment relative to the exopod of the female first swimming leg, the armature of the second and third endopodal segments of the second to fourth swimming legs, and the segmentation and chaetotaxy of the fifth swimming leg. The new species most resembles S. neglecta, primarily due to the loss of the inner seta on the first endopodal segment of the second to fourth swimming legs, and shape and setular ornamentation along the inner margin of the caudal rami. Due to the high diversity of copepods with marine ancestors in Vietnam, proximity to the sea has been suggested as a key factor influencing the diversity patterns of copepod fauna in the country.

The authors sincerely thank Dr Izwandy Idris, Dr Samuel Gómez, and an anonymous reviewer for their valuable revision of the manuscript.

Additional Information and Declarations

Competing Interests

Author Contributions

Data Availability

New Species Registration

The authors declare there are no competing interests.

Ngoc-Son Tran conceived and designed the experiments, performed the experiments, analyzed the data, prepared figures and/or tables, authored or reviewed drafts of the article, and approved the final draft.

Thi-Phuong Pham conceived and designed the experiments, performed the experiments, analyzed the data, authored or reviewed drafts of the article, and approved the final draft.

Minh Anh Dam conceived and designed the experiments, performed the experiments, prepared figures and/or tables, and approved the final draft.

Chaichat Boonyanusith conceived and designed the experiments, performed the experiments, analyzed the data, prepared figures and/or tables, authored or reviewed drafts of the article, and approved the final draft.

The following information was supplied regarding data availability:

The type materials are deposited at the Zoological Collection of Duy Tan University (ZC-105 DTU), Da Nang city, Vietnam.

The following information was supplied regarding the registration of a newly described species:

Publication LSID: urn:lsid:zoobank.org:pub:50CEF9AE-6C53-4366-BBD8-98ABE5E2F9FD

Schizopera vietnamica:urn:lsid:zoobank.org:act:FEC551E5-DBCA-4370-B0F7-347EDB0A2D76.

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
