# Peer review of "Schizoperavietnamica sp. nov. (Copepoda, Harpacticoida, Miraciidae), a new species from a mangrove zone in Vietnam, with description of a new species, and a key to the species from the Oriental region, Sulawesi, and East Asia"

_PeerJ, doi:10.7717/peerj.20246_

## Round 0.1 · original submission · Minor Revisions

Both reviewers commented on similar issues that need to be addressed by you, such as typographical errors, wrong labelling in figures and the use of correct terminology in the description of the species.

Another aspect that you should explain further is in the methodology section - how these specimens were collected. This is very important to allow comparison studies to be done in the future.

·

Basic reporting

The manuscript is well-written and structured, but I detected some minor typographical and grammatical errors. I am not an English native speaker, but took the liberty to suggest some corrections to improve the readability of the text. Also, I suggested some corrections that have to do more with the correct terminology used in the taxonomy of harpacticoid copepods. The authors cited the most useful literature for their investigation, but I suggest checking and including also "Apostolov, A. (1982). Genres et sous-genres nouveaux de la famille Diosaccidae Sars et Cylindropsyllidae Sars, Lang (Copepoda, Harpacticoidea). Acta Zoologica Bulgarica. 19: 37-42" since it contains useful information about the establishment of two subgenera of Schizopera that were not mentioned by the authors. The figures are of good quality and the tables are very informative, but I detected some errors in the numbering of the figures and their citation in the text. Finally, I suggested some corrections to the figure captions.

Experimental design

No comment.

Validity of the findings

No comment.

Additional comments

The authors presented a well-structured and illustrated description of a new species of the species-rich harpacticoid genus Schizopera from Vietnam. The manuscript is well-written and structured, but I detected some minor typographical and grammatical errors. I am not an English native speaker, but took the liberty to suggest some corrections to improve the readability of the text. Also, I suggested some corrections that have to do more with the correct terminology used in the taxonomy of harpacticoid copepods. The authors cited the most useful literature for their investigation, but I suggest checking and including also "Apostolov, A. (1982). Genres et sous-genres nouveaux de la famille Diosaccidae Sars et Cylindropsyllidae Sars, Lang (Copepoda, Harpacticoidea). Acta Zoologica Bulgarica. 19: 37-42" since it contains useful information about the establishment of two subgenera of Schizopera that were not mentioned by the authors. The figures are of good quality and the tables are very informative, but I detected some errors in the numbering of the figures and their citation in the text. Finally, I suggested some corrections to the figure captions. The authors added some value to their work by including a key to the species of Schizopera from the Oriental Region, Sulawesi, and East Asia.

Reviewer 2 ·

Basic reporting

No comment

Experimental design

No comment

Validity of the findings

No comment

Additional comments

The manuscript presents a comprehensive description of Schizopera vietnamica sp. nov. from Vietnam, and the authors should be applauded for their careful comparison with four other Schizopera species. Rigorous investigation performed to a high technical & ethical standard. In particular, the detailed observations on morphological differences such as the absence of the inner seta on both Exp-2 and Enp-1 of P2, P3, and P4 are noteworthy and strengthen the taxonomic justification of this newly identified species in Vietnam.
However, there are some aspects of the species description that appear to have been overlooked or insufficiently detailed. Although these issues are minimal, addressing them would improve the clarity and overall quality of the manuscript.

The following specific comments should be addressed:

Abstract
L25 : What is the mesh sized of plankton net used?

Materials & Methods
L85-87 :How was the samples collected? Was it vertically hauled air horizontally towed? If it was vertically hauled, what is the depth where the samples were taken?
L88 : Specify the model of the stereomicroscope.
L273 : 'P5 (Fig. 8B)' I think it should be Fig. 8C.
L273 : 'Exp suboval, as long as wide'. Please check this statement.
L295 : 'Some authors have defined the spine as an enlarged tubular pore (e.g., Karanovic 2006; Karanovic & McRae, 2013; Karanovic, Kim & Grygier, 2015)'. Echoing to Boonyanusith et al. statement in L269: 'the anterior surface of Exp-3 with enlarged tubular pore, representing hyaline spine swollen at base and remarkably tapering distally, inserted at the middle and close to inner margin, reaching distal 2/3 of distal half of Exp-3'. I do agree to just define the spine as 'enlarged tubular pore'.

Annotated reviews are not available for download in order to protect the identity of reviewers who chose to remain anonymous.

---

## Round 0.2 · accepted · Accept

All comments from reviewers are adequately addressed, and I think your manuscript is ready for publication (pending the technical guidelines from the journal).